# Transition from hospital to home care for preterm babies: A qualitative study of the experiences of caregivers in Uganda

Christine Nalwadda[1], Andrew K. Tusubira[1], Harriet Nambuya[2], Gertrude Namazzi[1], David Muwanguzi[3], Peter Waiswa[1], Jenny Kurinczuk[4], Maureen Kelley[5], Manisha Nair[4]*

1 Makerere University College of Health Sciences, School of Public Health, Makerere University, Kampala, Uganda, 2 Jinja Regional Referral Hospital, Jinja, Uganda, 3 Iganga District Hospital, Iganga, Uganda, 4 National Perinatal Epidemiology Unit, Nuffield Department of Population Health, University of Oxford, Oxford, United Kingdom, 5 The Ethox Centre and Wellcome Centre for Ethics and Humanities, Nuffield Department of Population Health, University of Oxford, Oxford, United Kingdom

☯ These authors contributed equally to this work.
* manisha.nair@npeu.ox.ac.uk

**Data Availability Statement:** We have made excerpts of the transcripts relevant to the study available within the paper and these can also be

## Abstract

Improving care for preterm babies could significantly increase child survival in low-and-middle income countries. However, attention has mainly focused on facility-based care with little emphasis on transition from hospital to home after discharge. Our aim was to understand the experiences of the transition process among caregivers of preterm infants in Uganda in order to improve support systems. A qualitative study among caregivers of preterm infants in Iganga and Jinja districts in eastern Uganda was conducted in June 2019 through February 2020, involving seven focus group discussions and five in-depth interviews. We used thematic-content analysis to identify emergent themes related to the transition process. We included 56 caregivers, mainly mothers and fathers, from a range of socio-demographic backgrounds. Four themes emerged: caregivers' experiences through the transition process from preparation in the hospital to providing care at home; appropriate communication; unmet information needs; and managing community expectations and perceptions. In addition, caregivers' views on 'peer-support' was explored. Caregivers' experiences, and their confidence and ability to provide care were related to preparation in the hospital after birth and until discharge, the information they received and the manner in which healthcare providers communicated. Healthcare workers were a trusted source of information while in the hospital, but there was no continuity of care after discharge which increased their fears and worries about the survival of their infant. They often felt confused, anxious and discouraged by the negative perceptions and expectations from the community. Fathers felt left-out as there was very little communication between them and the healthcare providers. Peer-support could enable a smooth transition from hospital to home care. Interventions to advance preterm care beyond the health facility through a well-supported transition from facility to home care are urgently required to improve health and survival of preterm infants in Uganda and other similar settings.

obtained by sending a request to data.access@ndph.ox.ac.uk. Information on how to access data is available on Oxford Population Health Department's website (https://www.ndph.ox.ac.uk/data-access). We are unable to make the whole transcripts available through a public repository or submit as supplementary files as it includes sensitive information from interviews of parents and caregivers of preterm infants. Even if we remove the names and other personal details, the data could be still potentially identifiable and can cause distress to participants. So, to protect confidentiality of the sensitive information we did not seek consent from the participants to share data in an open access repository. Participants consented to including quotes and excerpts of the transcripts in scientific reports, publications and presentations, but not to have entire transcripts made publicly available. Thus, making entire transcripts publicly available would breach compliance with the research protocol approved by the ethics committees, namely the Oxford Tropical Research Ethics Committee (OxTREC Ref 27-19), University of Oxford, the Institutional Review Board at Makerere University and the Uganda National Council for Science and Technology (Ref SS 5024).

**Funding:** The study was funded by a Nuffield Department of Population Health Pump-priming Award. MN is funded by a Medical Research Council (UK) Career Development Award (Ref:MR/P022030/1). The funders had no role in study design, data collection and analysis, decision to publish, or preparation of the manuscript.

**Competing interests:** The authors have declared that no competing interests exist.

## Introduction

Globally, about 15 million babies are born preterm each year, of which 81% are in Asia and sub-Saharan Africa [1]. A preterm birth is a baby born alive before 37 completed weeks of gestation [2]. Each year, preterm births are directly responsible for 1.1 million neonatal deaths worldwide [3]. This burden disproportionately affects poor and disadvantaged populations, particularly in low-and-middle income countries (LMICs) [4]. In Uganda, 14% (~200,000) of all infants born alive are born preterm and complications of prematurity account for about 38% of neonatal deaths (deaths within the first 28 days of birth) [1]. Strategies focusing on survival of preterm infants are therefore critical for the reduction of child mortality, particularly in LMICs.

While there are proven interventions to improve survival of preterm infants, most interventions focus on clinical/facility-based care during pregnancy, childbirth and immediately after birth [5]. There is little emphasis on support to provide adequate care at home after discharge from health facilities [6]. The majority of the preterm infants die due to a lack of simple essential care such as warmth and feeding support at home [7]. Despite evidence of high post-hospital discharge deaths of preterm infants in Uganda [8], there are no structures to support caregivers of preterm infants to continue recommended care practices at home (such as keeping the baby warm, appropriate feeding, infection control, and recognising early warning signs) [7]. There is also no support to overcome social pressures, stigma and other barriers in the community and family settings that affect survival of preterm infants [9]. Studies have discussed the issue of stigma related to small babies and preterm babies in sub-Saharan Africa, which can have a detrimental effect on the ability of the parents to care for the baby, as well as affect their own health and wellbeing [10]. For example, a study from Malawi found that in some communities, a premature baby is equated to having an abortion and was therefore perceived as "a curse" [11]. In a study in Ghana, stigma was linked to "undesirable" physical features by people who were unfamiliar with small babies [10].

In Uganda, the high burden of preterm births and their poor survival after hospital discharge requires integrated interventions to address the challenges of transition, continued care, stigma and misinformation. Although previous studies found that community health workers (CHWs) were able to help mothers to identify and refer sick newborns in rural Uganda, the process was challenging for preterm infants [12, 13]. There is evidence of usefulness of peer-support interventions delivered at the health facility or at the community to support families with preterm infants, mainly from high-income countries [14]. These include parents who have previously cared for preterm infants meeting with new parents face-to-face, or through telephone, or support groups that meet physically or online. Peer-support, which is identified within the construct of social relationships in healthcare provision, has been recommended and found to be effective in strengthening health promotion strategies [15]. An understanding of caregivers' needs and expectations through the transition process to care at home is essential to develop context specific support mechanisms/ interventions as we do not have evidence of effective interventions from LMICs. The aim of our study was to understand the challenges experienced by caregivers of preterm infants in Uganda while transitioning from hospital to home care in order to identify the needs, opportunities and potential interventions for improving support systems.

## Methods

### Study design

We conducted a qualitative study from June 2019 to February 2020 using focus group discussions (FGDs) and in-depth interviews (IDIs) with caregivers of preterm infants to explore

their experiences, expectations, challenges and available support systems for transitioning from hospital to home care. While FGDs captured shared experiences, the IDIs elicited lived experiences of the caregivers who did not want to participate in a group discussion.

## Study setting

The study was conducted in Busoga region, in eastern Uganda, in the districts of Iganga and Jinja. Busoga is one of the poorest regions with some of the poorest health indicators in Uganda. The region's post-discharge death rate among preterm infants ranges from 11.3% to 28% [16], compared with the national estimate of 14% [1]. Iganga district has a population of about 500,000, 95% of whom live in rural areas with poor access to healthcare. In contrast, Jinja district is an industrial city with nearly two-thirds (63.3%) of its 501,300 population living in urban areas. In Iganga, comprehensive preterm care, which includes all essential newborn care as well as advice to mothers at discharge for caring for the preterm baby at home, is provided at Iganga general hospital while in Jinja this care is provided at the regional referral hospital, which caters to the population of Jinja city and 11 districts in the South-Eastern region of Uganda.

## Study population

Our study population included caregivers of preterm infants defined as parents (mothers and fathers), or other relatives or carers. We included caregivers with a preterm infant born within 1–12 months prior to data collection. We did not include caregivers whose infants were critically ill, i.e. infants who had life-threatening complications, during the study period and those who did not provide informed consent.

## Sample size and sampling method

We conducted seven FGDs (each with 6–8 participants) and five IDIs among a purposively selected sample of caregivers of preterm infants. The final sample size was informed by thematic saturation. Our sampling strategy was informed by maximum variation, which entailed selection of participants with a range of household demographics including rural-urban residence and marital status; preterm status (extreme preterm to near full term); and considering both gender and age range of the participants. Although our approach was maximum variation, we did not aim to recruit equal numbers of mothers, fathers, relatives or participants from different demographic and socioeconomic status, which would make the study extensive and not practically feasible. Participant recruitment was guided by the staff in charge of the neonatal care units at the study hospitals who also trained the study staff on how to approach caregivers of preterm infants. The hospital staff provided a list of eligible participants with contact details and were not directly involved in the recruitment of the participants to avoid selection bias. The study team selected and approached the participants from the list using purposive sampling to attain maximum variation, as described above.

## Data collection

Prior to data collection, we pretested a focus-group discussion guide among caregivers with preterm infants from different localities within Iganga district, who were selected from the hospital discharge records. The pre-tests were conducted in a manner that replicated the anticipated flow of data collection sessions, including administration of consent forms, demographic questionnaire, and discussion guide. We then revised the study materials to ensure that questions were clear, appropriate, and aligned with the overarching research question. The pre-tested guides included the key topics to explore caregivers' experiences before, during

and after childbirth. It included questions on experiences while in the hospital after birth, during discharge and when back at home with the preterm infant, information and support received at the facility and in the community. Probes were included to explore social and economic challenges associated with home care of preterm infants, ability and challenges of caregivers to understand and follow the recommended care, barriers and facilitators to seeking appropriate healthcare, and community and cultural attitudes towards preterm infants. We also included a specific question to explore caregivers' views on 'peer-support'. The guides were translated to "Lusoga", the language most commonly spoken in the Busoga region, and back-translated to English. The focus-group guide was also used for conducting in-depth interviews with caregivers who were not comfortable in participating in group discussions.

Three (one male and two female) trained qualitative researchers conducted the FGDs and interviewed caregivers using the discussion guide. The discussions and interviews were conducted in person in the local language, Lusoga. On an average, the FGDs and IDIs lasted for about 45 minutes to an hour. All discussions and interviews were audio recorded and later transcribed verbatim in Lusoga and translated to English. During coding and analysis, we discussed issues of meaning related to translation. Field notes and observations were taken by a support researcher (note-taker) during the group discussions. All transcripts were proofread against original recordings to check for translation accuracy and cross-checked with field notes.

## Data analysis

We used a thematic content and framework analysis approach to allow new insights and inductive categories to emerge directly from the qualitative data [17]. Content analysis interprets meanings from the text (data), a concept known as the natural paradigm. With this approach, codes are derived directly from text data. We modified this approach, including some codes informed by the research questions and discussion guides, which were grounded in the literature review [18]. An iterative team coding approach was used to check interpretations and improve validity, including discussion of cross-cultural concepts and meaning [19]. A team of five researchers (with two researchers external to the data), each read through two transcripts and conducted an initial coding of the 12 transcripts. The team then discussed coded scripts, making revisions once agreed. A codebook was developed and further refined following a second round of each researcher coding two transcripts. Through consensus, the codebook was revised and a primary code list applied to all transcripts using ATALAS.TI.9, while allowing for new codes. A secondary coder checked and made revisions, noting any disagreement. The full team conducted a tertiary review of the coded transcripts and then compared and discussed the findings to generate cross-cutting themes across the transition process from preparation in the health facility to caring for the infant at home.

## Ethical approval

The study was approved by the Oxford Tropical Research Ethics Committee (OxTREC Ref 27–19), University of Oxford, the Institutional Review Board at Makerere University and the Uganda National Council for Science and Technology (Ref SS 5024). Administrative clearance was also obtained from each district health office and also from each health facility in Iganga and Jinja districts. Written informed consent was obtained from all participants.

## Results

A total of 56 caregivers participated in the study, 51 in the FGDs and five in IDIs. Most participants were female (n = 41) and the majority belonged to the Basoga tribe (n = 44). Twenty nine out of the 56 participants were between 25 and 34 years of age, 80% (n = 45) were married

and more than half (n = 33) had secondary or tertiary level education. A description of the study population is presented in Table 1.

Four themes emerged from the FGDs and the IDIs: caregivers' experiences through the transition process from preparation in the hospital to providing care at home; appropriate communication; unmet information needs; and managing community expectations and perceptions. Participants' views on peer-support are reported as a separate theme, although it was a key discussion point within each of the four emergent themes.

## Caregivers' experiences through the transition process

Caregivers experienced fear, worry and confusion whilst in the health facility and while caring for the infant at home after discharge. Although they largely felt supported by healthcare workers while in the hospital, there were questions about appropriate communication. There was, however, a lack of adequate support for caregivers to care for the baby at home after being discharged from the health facility and they received mixed, and sometimes inaccurate, information from different sources.

## Preparation in the health facility

Caring for a preterm infant after birth was a new and challenging experience for caregivers, which in many instances, was described as confusing. Some parents stated that they feared

**Table 1. Sociodemographic characteristics of the participants.**

| Variable | Frequency (n = 56) | Percent |
|---|---|---|
| **Gender** | | |
| Female | 41 | 73.2 |
| Male | 15 | 26.8 |
| **Tribe** | | |
| Basoga | 44 | 78.6 |
| Other tribe | 12 | 21.4 |
| **Caregivers' age (in years)** | | |
| 18 to 24 | 18 | 32.1 |
| 25 to 34 | 29 | 51.8 |
| 35 to 45 | 9 | 16.1 |
| **Age of infants in months (n = 42*)** | | |
| 3 to 5 | 30 | 71.4 |
| 6 to 11 | 12 | 28.6 |
| **Marital status** | | |
| Single | 11 | 19.6 |
| Married | 45 | 80.4 |
| **Education level** | | |
| No education | 2 | 3.6 |
| Primary | 21 | 37.5 |
| Secondary | 30 | 53.6 |
| Tertiary | 3 | 5.4 |
| **Caregivers of the preterm babies** | | |
| Mothers | 40 | 71.4 |
| Fathers | 15 | 26.8 |
| Non-parental (Female) caregiver | 1 | 1.8 |

*For some infants, both parents participated in the FGDs so the total does not add up to 56.

their newly born preterm baby because they looked "small" and fragile. This could be related to the lack of preparedness from the start of pregnancy due to a lack of information about how to care for a preterm infant. In addition, some mothers did not know that they were having a preterm birth due to incorrect diagnosis and others could not accept or "believe" that they were going to give birth before term even when they were told.

> "*I didn't know I was going to give birth before time. They [healthcare providers] did not tell me until the last minute. Even at the time of delivery, they were seeing one baby. Yes, I used to go for antenatal [care] and I went four times. Even the doctor here used to check me and he only told me that I have one baby and it is very small. . . . but I gave birth to twins at seven months*" (P5, Mother, FGD).

> "*It is doctor. . . . who monitored my pregnancy . . . he told me your uterus is weak, it will not be able to carry the baby up to nine months, you will not push but they will operate you at eight months and a half, but I couldn't believe it.*" (P8, Mother, FGD).

While some feared touching their babies, others recognised this as a new challenge and accepted the need to learn how to care for their baby from the healthcare providers. A mother narrated the challenges in providing Kangaroo Mother Care (KMC), which is skin-to-skin contact to keep the infant warm. She was worried about harming or dropping the *'small'* baby while carrying him around.

> "*Kangaroo was difficult because the baby was small so it gave me a bit of hard time doing it . . . our babies were very tiny and they even scare us who produced them to the extent that we even don't want to carry them*" (P3, Mother, FGD).

Mothers frequently reported getting support from healthcare providers, mainly hands-on training on KMC, breastmilk expression and feeding. However, in some instances, they felt intimidated and at times overwhelmed by the instructions given by the healthcare workers. This is discussed further under the 'appropriate communication' theme.

## Care at home after being discharged from the health facility

Once parents left the hospital, there was no clear path for continued support at home offered by the healthcare providers. Caregivers shared that one scheduled visit to the hospital was too little and not soon enough as the caregivers experienced more challenges in caring for the preterm infants in the first few weeks after discharge. At home, caregivers were not always aware of the baby's needs, danger signs and where to seek help and treatment when the need arose. Due to their inability to notice danger signs in the baby, caregivers sometimes did not respond by seeking timely advice and care.

> "*I would fear because when you are here in the hospital you know the health worker is near, just in case the child gets a problem the health worker will help you but at home you are alone and it took me a long time to get comfortable. . . In the first weeks I wished I remained in the hospital where there are health workers*" (P1, Mother, IDI).

Mothers expressed the need for continued care and support from healthcare workers or *'counsellors'* who could visit caregivers at regular intervals after discharge from the hospital to assess hygiene and the overall care environment for the infant at home. This would have encouraged and reassured them and also improve their confidence in caregiving.

Caregivers raised worries about the extra costs of care while in hospital and in transitioning to home after discharge. They found it difficult and costly to purchase medicines, by which the caregivers meant essential micronutrient supplements and prophylaxis for anaemia in preterm infants such as multivitamin syrup and haematinics, and worried that this would compromise the health of the baby.

"*Buying medicine was very expensive for us. The medicines only last two weeks and another type lasts only one week and they get finished. So, it was a bit expensive.. . . We try to save some money for the medicines*" (P1, Mother, IDI).

## Appropriate communication

While in the health facility, staff communication with mothers was important and had a strong influence on their preparedness. Although most mothers felt supported by healthcare providers, some experienced improper communication and lack of empathy from the healthcare providers. For example, some felt that healthcare providers were impatient or rude while giving instructions, which discouraged new mothers of preterm infants who needed more time and support to learn to provide KMC and express breastmilk or breastfeed. One mother expressed her disappointment with healthcare workers when her infant was in a neonatal intensive care unit (NICU):

"*When my baby was in an incubator the healthcare workers provided a lot of care . . . but they were rude. As you know the way they speak, but I will not say they did not give us care. They cared for us but that rudeness. Some of us we have high blood pressure. They could buck [shout] at you and even the pressure rises. . .*" (P2, Mothers, FGD).

Mothers did not feel entirely trusted by healthcare providers to care for the baby after leaving the hospital. Healthcare providers made the mothers practice KMC under strict supervision in the hospital by instigating fear about the need to re-hospitalise the infant after discharge due to their perceived lack of competence.

*When they were discharging me,. . ., they [healthcare providers] told us that you are going but we know you are ending back here [if you don't put] those infants on the chests. They told us to rehearse everything. We [mothers] could go to the health worker with our baby and they demonstrated very well how to wrap the baby in the chest* (P3, Mother, FGD).

Fathers, on the other hand, felt left-out from the start as there was very little communication with them and were provided limited or no information. They felt less supported and less informed by healthcare providers about continuity of care of the baby at home after discharge. Consequently, this affected their involvement in caring as they felt less informed and empowered.

"*. . . the reason why they [male partners/husbands] do not do it [provide care to the preterm baby], is that when they [health workers] are discharging the woman from the health facility you will not be there so you will not know the instructions they have given her. The health worker may not take the responsibility to tell you when you are going back. So usually when they are discharging, always the man will be busy, you will be arranging transport to take these people home and you will not know what the health worker will instruct you to do.*" (P3, Father, FGD)

Lack of communication with fathers also meant that the burden of care was on the mother. It was suggested by one father that both men and women should be sensitised about preterm birth in the community and given adequate information to enable them to prepare and cope in case they have a preterm infant.

"*So if they bring such sensitizations in the community and sensitize all people including men. Men are then told how to treat their wives when pregnant, the responsibility of men and the responsibilities for women when they have preterm babies. But with us here, when the sensitization sessions come in the community, we leave them for women.*" (P4, Father, IDI)

## Unmet information needs

Healthcare workers were a trusted source of information for care of preterm infants while in the hospital after birth. They were also expected to prepare mothers to withstand the challenges that they could face in caring for the infant at home after being discharged from the hospital.

"*They [healthcare workers] should encourage these mothers because in the communities where we live people have negative attitudes towards having premature infants so they should put more efforts to encourage them [mothers] that even premature infants are human beings. When mothers produce premature infants, they get disgusted and they hate themselves and even when they look at the baby they wish if the baby died. They see it as something unusual so they [healthcare workers] should put more effort to encourage mothers . . . . . . that the baby will grow like any other child*" (P4, Mother, FGD).

While at the facility, the type of information caregivers received from healthcare providers were mainly related to KMC, breastfeeding, hygiene, protecting the infant from infection, regularly giving vitamins and supplements. However, information varied in terms of frequency of KMC and breastfeeding. In addition, some caregivers ignored the information, falling back on practices that were more familiar in the home or community.

"*. . . the doctor told me not to touch the baby when I have smeared Vaseline. He told me to forget about Vaseline. I took time without using Vaseline and the child had no problem. But, at one point I thought what happens when you smear Vaseline? So, I smeared Vaseline and I tied the baby in the chest and the baby sneezed and sneezed and sneezed, it had difficulty in breathing and started producing foam through the nose, the baby started convulsing and I was alarmed, . . .I called the healthcare worker.*" (P6, Mother, FGD).

After discharge from the hospital, navigating through healthcare facilities was confusing as caregivers were allowed only a single visit back to the hospital for advice and support and thereafter were referred to other facilities. Consequently, due to a lack of proper information, they described moving across many facilities to get treatment for any perceived danger sign. In some cases, caregivers did not receive the kind of care and welcome at these health facilities compared to the hospital where they gave birth, which negatively affected care seeking.

"*When I called the health worker, they told me that those are now old infants take them to . . .[another facility]. When I reached . . . [the other facility], there were almost no health workers to help me. After pleading, the baby was treated, but very late, at 10pm. Got treatment only after quarreling with the health workers and I wanted to just escape from there and go to a private facility.*" (P2, Father, FGD).

## Managing community expectations and perceptions

Many caregivers were distraught by the negative perceptions of members of their community who raised doubts about the development and survival of the infant. They felt that no one at home or in the community understood that although preterm infants needed more care, they could develop and grow normally.

> "*My neighbour told me that children who are born premature are always dull in class because their brains are not mature enough, that they do very badly in class, and she said you wait and see those ones, the period which the baby would have started to crawl they will just start sitting, the time when they will have started standing it is when they will start crawling*" (P4, Mother, FGD).

Caregivers shared that community members' attitudes about preterm infants affected their experiences in caring for their babies. Most community and family members expect to see and hold the baby shortly after birth, but when caregivers refused to '*show*' their child/children to protect the infant/s from infection as advised by the healthcare providers, they were deemed as being arrogant. One mother had to close her business for two weeks because the customers wanted to see and hold the twins, and when she refused, they were unhappy. There was agreement among the participants that people (neighbours, friends and extended family) often instigate fear and stigma around preterm infants.

> "*I gave birth at seven months but the baby was very tiny and people said that is a rat, you just go and throw it away, but I told them that this is a baby, but they told me you go and throw that baby in a dust bin, but I told them that I cannot throw my baby in dust bin.*" (P9, Mother, FGD)

> "*They say in our clan we don't produce such babies. So when they tell you in their clan they don't produce those so it is you in your clan which produces, it is a bit challenging.*" (P5 and P8, Mothers, FGD)

Caregivers also received incorrect advice and pressure from community members. They confused the caregivers by giving advice that conflicted with what they were taught in the hospital.

> "*Those health workers deceived them. Such babies are kept near hot Jerrycans, and they are put near charcoal stove, so that the baby can grow well. But for you, you brag that they kept the babies in the incubator! But I refused to do what they [community members] were telling me to do. . . ., I became firm. I could lock my babies inside the house and I could not show them my babies. They only saw them when they have grown up.*" (P7, Mother, FGD)

## Peer-support

Peer-support was related to each of the above four themes as it was considered to have the potential to address caregivers' information needs, bridge the communication gap between healthcare providers and caregivers (mothers and fathers) during the process of discharge, improve caregivers' confidence in providing care at home, improve their ability to access appropriate health care, provide reassurance to them and deal with the negative attitudes and perceptions of the community.

Mothers reported being encouraged by their peer-preterm caregivers (other mothers who gave birth to a preterm infant) while in the health facility. In some cases, peer-preterm caregivers helped the mothers to provide KMC and to express breast milk.

*"What gave me courage was putting us in one place all of us with preterm babies. . . . We used to support each other during kangaroo [KMC]."* (P1, Mother, IDI)

*". . . . . . whenever you could look at your colleague [other mothers of preterm babies] you could gain courage."* (P5; Mothers, FGD)

Acknowledging the difficulty in receiving continued care from healthcare providers after discharge from the hospital, several mothers volunteered to share the knowledge and experience that they gained through trial and error to help others who have a preterm birth so that they do not go through the same challenges.

*"..so those people [from the community] are supposed to support us but they instead discourage us. So I want to encourage my fellow mothers that these premature babies grow but those people out there they discourage us that the baby doesn't grow, that a premature baby must die, but these babies survive and they should see our babies as an example that these babies survive and they grow."* (P6, Mother, FGD)

While the fathers felt *'left-out'* due to lack of adequate communication and information from healthcare providers, they felt supported by other fathers who had a preterm child in the hospital, especially through information sharing.

*". . . even myself when I got a premature [baby] I got very worried because I had never seen them, but the problem is that most of us it gets us unaware and we don't know anything about premature babies and other people think that premature babies don't survive, because a few of them survive. Because me when I was in Kibuli we were very many people with premature babies, and we shared phone contacts."* (P2, Father, FGD)

Examples were shared about HIV programmes in which peer-support enabled HIV positive people to share their fears and worries, and to obtain information from others who had the infection and were undergoing treatment.

*"We [men] can get involved [in care] if we are informed, because if someone gets such a baby it is easy to tell that person there is even this person who has experienced that, they can call that person who can tell you this and this and this and this, the way you see people with HIV, they can call someone who testifies that I also have it [HIV infection] but I am on medication and you can also take medicine to improve your life"* (P4, Father, FGD)

Although there was a general consensus that support from other parents (both mothers and fathers) who had experience of caring for a preterm infant would increase caregivers' confidence and ability to provide care, it was felt that getting such support might be difficult due to the stigma associated with preterm birth.

*"It [peer-support] is a good idea only if people are willing to open up because it is not many people who will come out and disclose and tell you that I produced a premature baby. Some of us have stigma, we fear to be talked about that this one produced a premature because some people turn it into a song, but it [peer-support] is not a bad idea."* (P2, Mother, FGD)

**Table 2. Summary of the themes, sub-themes and key points from the qualitative research.**

| Themes | Sub-themes | Key points |
|---|---|---|
| Caregivers' experiences through the transition process | Preparation in the health facility | Fear, worry, confusion among parents/ caregivers.<br>Adequate support from healthcare providers to mothers, but improper communication. |
| | Care at home after being discharged from the health facility | No continued support at home for parents/ caregivers.<br>Caregivers not aware of baby's needs.<br>High costs of supplements and medicines. |
| Appropriate communication | -- | Improper communication from healthcare providers.<br>Lack of empathy from hospital staff.<br>Mothers did not feel trusted by hospital staff.<br>Fathers felt left-out. |
| Unmet information needs | -- | Healthcare workers are a trusted source of information while in the hospital.<br>Parents/ caregivers confused after discharge from the hospital.<br>Lack of proper information for parents/ caregivers after discharge. |
| Managing community expectations and perceptions | -- | Negative perceptions of community members about preterm babies.<br>Lack of understanding about preterm care and support among community members.<br>Community members instigating fear and stigma in relation to preterm babies.<br>Incorrect advice and pressure from community members in relation to caring for the preterm babies. |
| Peer-support | -- | Potential to address unmet information needs and communication gap, and facilitate continued support after discharge.<br>Mothers felt encouraged by peer-supporters.<br>Fathers felt supported by other fathers of preterm infants in the hospital.<br>Mothers volunteered to be peer-supporters.<br>Could be difficult to find peer-supporters due to stigma related to preterm birth. |

A summary of the themes and sub-themes with key points are presented in Table 2.

## Discussion

Findings from our study revealed the complexities and challenges related to transition of care of preterm infants from hospital to home. Caregivers' experiences and their confidence and ability to care for their infant were related to preparation in the hospital after birth and until discharge, the information they received and the manner in which healthcare providers communicated with them. Healthcare workers were a trusted source of information while in the hospital, but there was no continuity of care after discharge which increased caregivers' fears and worries about the survival of their baby. They often felt confused, anxious and discouraged by the negative perceptions and expectations from the community. Fathers felt completely left-out as there was very little communication between them and the healthcare providers while in the hospital, no information was provided at discharge to involve them, and therefore they felt incapable of sharing care responsibilities with the mother at home. It appeared from the discussions that peer-support (from mothers and fathers) while in the hospital could complement the support from healthcare providers in terms of providing courage, empathy, reassurance and confidence to the caregivers. Ideas were shared about organised peer-support but this could be challenging to implement due to the stigma around preterm birth.

The challenges related to transition of care from hospital to home for preterm infants have been discussed in other studies, but the majority are from high income settings [20]. A qualitative study from Australia that interviewed 40 parents of preterm infants to explore their perception of readiness before and after discharge from NICU found information and communication to be a major factor [21]. Parents did not feel that they received adequate information about the discharge process and at times felt overwhelmed and rushed. Healthcare

providers' rudeness and lack of respect while delivering pregnancy and child health services have been highlighted in other literature [22–24], although not specifically related to the transition process for preterm infants. Nevertheless, hospital staff taught the parents to care for the infant at home and most felt that they were prepared, but some parents were not mentally or physically prepared and therefore felt anxious.

A study from China found the quality of teaching and content of information provided to parents during discharge to be an important determinant of parents' readiness [25]. Similar to our study, parents' frustration due to inconsistent and conflicting information from healthcare providers was a theme that emerged from other research [21]. There are several studies that have highlighted aspects of stress, worry and confusion among parents through the complex transition process [26, 27] and adjusting to home care [28]. Fear and worry about the care and survival of the infant without support from healthcare providers at home was a theme that was identified in a meta-synthesis of twelve qualitative studies on transition from hospital to home care of preterm infants [20].

A literature review of fifty studies published between 1980 and 2014 found that caregivers needed both professional and social support during the process of transitioning from hospital to home [29]. A network of support in the community was essential for continuity of care and adequate information [20]. Reassurance from healthcare providers in baby clinics and support from friends and family were identified as key determinants of mothers' improved ability and confidence in caring for her baby [20, 30]. Most mothers found it hard to adjust and felt the need for regular support during the first week at home after discharge [20]. A majority of the studies were again from high income countries where post-discharge support from health visitors and community nurses was available for caregivers [20]. In our study, mothers expressed the need for "counsellors" in the community who could provide continuity of care. However, such support systems are not available in Uganda and many other LMICs where caregivers are mostly left to learn by 'trial and error' [31], resulting in high rates of post-discharge deaths of preterm infants. Thus, as identified in our study, in this context the role of peer-support and other measures to ensure continuity of communication and care could be vital.

A recent study from Ghana exploring the support needs of parents of preterm infants while caring for the baby at home found that women relied more on friends and extended family, as access to continued support from healthcare workers becomes difficult after discharge from the hospital [6]. Caregivers in our study shared the challenges and worries related to the extra costs associated with buying essential supplements for the preterm infant, which are not included in the Ugandan national list for essential medicines, and thus not provided free of cost. These costs can be a significant financial shock to a family already living in an economically precarious situation. Parents receiving continued care at home looked for material support from other parents, which was also identified by other studies [6, 20, 32]. Such health system challenges will be a critical consideration in any guidance for transitioning to care at home.

A systematic review of seventy-two published studies found that structured and organised peer-support interventions, either face-to-face or telephone-based or internet-based, were effective in improving communication and parents' ability to care, but communication should continue throughout the process of preparing the parents for discharge to care at home [14]. However, while there are several studies evaluating the benefits of peer-to-peer support for parents in the NICU [32–37], evidence about the effectiveness of such support at home is limited [6]. In the context of Uganda, continuity of care through organised peer-support for both mothers and fathers starting at the hospital as part of the discharge process and continuing at home could be an effective support mechanism, but may also require support from community health workers. Although evidence about effectiveness of such support for preterm infants

is limited [6], in the context of adolescents with HIV in Uganda, peer-to-peer support has been found to be feasible, trusted and beneficial in supporting, motivating adherence to treatment, and coping with stigma among adolescents living with HIV [38] and is recommended by the World Health Organisation [39]. Therefore, peer-support could be an effective intervention for improving the transition from hospital and continuity of care at home for preterm infants in Uganda, if designed and implemented taking into consideration the contextual needs.

The requirement for involving the father from the beginning was identified in our study. A systematic review of fourteen qualitative studies explored fathers' experience and found lack of involvement in care from the start, and lack of information and communication from healthcare providers were important challenges to their engagement in care [40]. Fathers feeling 'left out' and insecure, and taking much longer to bond with their baby and learning to provide adequate care have been previously identified [20].

Similar to our findings, the community's perception of preterm infants, including myths and taboos about small babies, and strong views about how to provide care were identified as barriers to adequate care of preterm infants at home in a study from Indonesia [30]. Thus, in addition to support for caregivers, it is important to improve community awareness about preterm births through appropriate public health messaging as community's perception and curiosity about the preterm infant are known to adversely influence parents' experiences of care [20].

## Strengths and limitations

This is the first study that we know of which explored the experiences of caregivers of preterm infants related to the transition process from hospital to home in Uganda. It includes information shared by 56 participants through FGDs and IDIs and highlights the experiences of both mothers and fathers from a range of socio-demographic backgrounds. However, a limitation of the study is that lived experiences of parents whose infants died were not included. We could not interview caregivers of preterm infants who died at home after discharge from the hospital due to cultural sensitivities. It is likely that their experiences and perceptions could be different from parents whose infants were alive at the time of interview. We were also unable to recruit caregivers with very young infants, aged 1–2 months, who were unable to spare time for participating in the study. Another limitation was that we did not capture the perspectives of healthcare providers about the preparation in the hospital and the discharge process, and continued care provision and support at home.

## Conclusion

Bringing a preterm baby home has special challenges. Interventions to advance preterm care beyond the facility through a well-supported transition from facility to home care are important to improve the health and survival of preterm infants in Ugandan settings. The communication and information void needs to be addressed starting from the preparation in the hospital. Actively engaging fathers from the time of birth of the preterm infant could be valuable as many are keen to be involved in caring for their baby, but feel left-out and are not given adequate information at the health facility. Time limitation is a major component in terms of care provision by healthcare workers and reinforcement by others, possibly a peer-supporter along with a community health worker could complement the care and information provided by professionals. In addition, essential medicines and supplements for preterm infants could be included in the national list of essential medicines so that they are available for free from the hospital even after the infant is discharged. Thus, organised continued support

programmes will need to be carefully planned and evaluated along with parents, other caregivers, healthcare workers, and family and community members focussing on how the programme is delivered, its content and the environment to ensure acceptance and thus effectiveness.

## Supporting information

**S1 File. PLOS' questionnaire on inclusivity in global research.**
(DOCX)

## Acknowledgments

We thank all caregivers of preterm infants who participated in the study and shared their experiences.

## Author Contributions

**Conceptualization:** Christine Nalwadda, Jenny Kurinczuk, Maureen Kelley, Manisha Nair.

**Data curation:** Christine Nalwadda, Andrew K. Tusubira, Harriet Nambuya, Gertrude Namazzi.

**Formal analysis:** Christine Nalwadda, Andrew K. Tusubira, Gertrude Namazzi, Jenny Kurinczuk, Maureen Kelley, Manisha Nair.

**Funding acquisition:** Jenny Kurinczuk, Maureen Kelley, Manisha Nair.

**Investigation:** Christine Nalwadda, Maureen Kelley, Manisha Nair.

**Methodology:** Christine Nalwadda, David Muwanguzi, Jenny Kurinczuk, Maureen Kelley, Manisha Nair.

**Project administration:** Christine Nalwadda, Manisha Nair.

**Resources:** Manisha Nair.

**Supervision:** Christine Nalwadda, Harriet Nambuya, David Muwanguzi, Peter Waiswa, Manisha Nair.

**Writing – original draft:** Christine Nalwadda, Andrew K. Tusubira, Manisha Nair.

**Writing – review & editing:** Harriet Nambuya, Gertrude Namazzi, David Muwanguzi, Peter Waiswa, Jenny Kurinczuk, Maureen Kelley.

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
