## [Decision Letter · Decision Letter 0]

29 Sep 2022

PGPH-D-22-00378

Transition from hospital to home care for preterm babies: a qualitative study of the experiences of caregivers in Uganda

Dear Dr. Nair,

Thank you for submitting your manuscript to PLOS Global Public Health. After careful consideration, we feel that it has merit but does not fully meet PLOS Global Public Health’s publication criteria as it currently stands. Therefore, we invite you to submit a revised version of the manuscript that addresses the points raised during the review process.

Your manuscript has been assessed by an expert reviewer, whose comments are appended below. The reviewer, while broadly positive about your manuscript, has highlighted areas where the presentation of results and limitations could be improved. Please ensure you respond to each point carefully in your response to reviewers document, and modify your manuscript accordingly.

Please note that we have only been able to secure a single reviewer to assess your manuscript. We are issuing a decision on your manuscript at this point to prevent further delays in the evaluation of your manuscript. Please be aware that the editor who handles your revised manuscript might find it necessary to invite additional reviewers to assess this work once the revised manuscript is submitted. However, we will aim to proceed on the basis of this single review if possible.

We look forward to receiving your revised manuscript.

Kind regards,

Joseph Donlan

Staff Editor

Journal Requirements:

Additional Editor Comments (if provided):

Reviewers' comments:

Reviewer's Responses to Questions

**Comments to the Author**

1. Does this manuscript meet PLOS Global Public Health’s publication criteria? Is the manuscript technically sound, and do the data support the conclusions? The manuscript must describe methodologically and ethically rigorous research with conclusions that are appropriately drawn based on the data presented.

Reviewer #1: Yes

2. Has the statistical analysis been performed appropriately and rigorously?

Reviewer #1: N/A

3. Have the authors made all data underlying the findings in their manuscript fully available (please refer to the Data Availability Statement at the start of the manuscript PDF file)?

Reviewer #1: Yes

4. Is the manuscript presented in an intelligible fashion and written in standard English?

Reviewer #1: Yes

5. Review Comments to the Author

Reviewer #1: Congratulations to the authors for a beautiful manuscript! This manuscript describes the results of a qualitative study of 56 caregivers of preterm infants in Uganda through 7 FDDs and 5 IDIs. This manuscript is very well-written and provides excellent information about the the challenges experienced by caregivers of preterm infants in Uganda while transitioning from hospital to home care in order to identify the needs, opportunities and potential interventions for improving support systems. The introduction is well organized; the methods and analysis section is very clear and complete. I have only a few comments that could strengthen the results and discussion sections:

Results:

Page 9, line 184-191

I am not sure the quotes fully support and/or link the statement "their newly born preterm baby because they looked “small” and fragile." to lack of acceptance or denial among parents. The quotes don't really convey the lack of acceptance or denial among parents. Just because the parents expressed fear, it doesn't mean that they are denying their preterm infant. Perhaps the challenge here is to distinguish the pregnancy/birthing process from the actual infant?

Discussion

Page 20, line 446-448: Either here or in the relevant place in the results, it would be interesting to include specific drugs or medicines that were identified by parents as being costly to procure and that weren't provided free of charge for the health system.

Rudeness of the providers is documented in the literature although not specifically when related to caring for preterm infants. It could strengthen the discussion section to include one or two of such articles and related points.

Link the comments about "counsellors" on page 11, line 223 to the discussion about peer support more directly.

Page 21, line 480-482: I don't agree with the claim made by the authors that they included perspectives of caregivers whose infants had died by interviewing a parent of twins where one of the twins had died. Because the experience of having a preterm infant who died is significantly different, I suggest revising this statement to say that a limitation of the study is that the lived experience of parents whose infants died were not included.

6. PLOS authors have the option to publish the peer review history of their article (what does this mean?). If published, this will include your full peer review and any attached files.

**Do you want your identity to be public for this peer review?** For information about this choice, including consent withdrawal, please see our Privacy Policy.

Reviewer #1: No

---

## [Decision Letter · Decision Letter 1]

14 Feb 2023

PGPH-D-22-00378R1

Transition from hospital to home care for preterm babies: a qualitative study of the experiences of caregivers in Uganda

Dear Dr. Nair,

Thank you for submitting your manuscript to PLOS Global Public Health. After careful consideration, we feel that it has merit but does not fully meet PLOS Global Public Health’s publication criteria as it currently stands. Therefore, we invite you to submit a revised version of the manuscript that addresses the points raised during the review process.

The manuscript has undergone a thorough evaluation by four experts in the field, and their feedback is presented below. The reviewers have suggested incorporating additional information regarding the context to the care systems that is currently adopted in the country where the study is conducted They have also asked for more specific details regarding the qualitative study methodology, including participant selection, the procedures for conducting focus groups, and the gender representation within the focus groups. We recommend following the COREQ guidelines for reporting on qualitative studies to ensure the clarity and transparency of the methodology.

Could you please revise the manuscript to carefully address the concerns raised?

We look forward to receiving your revised manuscript.

Kind regards,

Lucinda Shen, MSc

PLOS Staff Editor

Journal Requirements:

Additional Editor Comments (if provided):

Reviewers' comments:

Reviewer's Responses to Questions

**Comments to the Author**

1. If the authors have adequately addressed your comments raised in a previous round of review and you feel that this manuscript is now acceptable for publication, you may indicate that here to bypass the “Comments to the Author” section, enter your conflict of interest statement in the “Confidential to Editor” section, and submit your "Accept" recommendation.

Reviewer #1: All comments have been addressed

Reviewer #2: (No Response)

Reviewer #3: (No Response)

Reviewer #4: All comments have been addressed

2. Does this manuscript meet PLOS Global Public Health’s publication criteria? Is the manuscript technically sound, and do the data support the conclusions? The manuscript must describe methodologically and ethically rigorous research with conclusions that are appropriately drawn based on the data presented.

Reviewer #1: Yes

Reviewer #2: Yes

Reviewer #3: Yes

Reviewer #4: Partly

3. Has the statistical analysis been performed appropriately and rigorously?

Reviewer #1: N/A

Reviewer #2: N/A

Reviewer #3: N/A

Reviewer #4: N/A

4. Have the authors made all data underlying the findings in their manuscript fully available (please refer to the Data Availability Statement at the start of the manuscript PDF file)?

Reviewer #1: Yes

Reviewer #2: No

Reviewer #3: Yes

Reviewer #4: Yes

5. Is the manuscript presented in an intelligible fashion and written in standard English?

Reviewer #1: Yes

Reviewer #2: Yes

Reviewer #3: Yes

Reviewer #4: Yes

6. Review Comments to the Author

Reviewer #1: (No Response)

Reviewer #2: Dear Editor,

This submission presents careful qualitative research on the experience of people with the subject of preterm babies. I believe it makes a useful contribution. The English is good overall; however, the manuscript still requires some improvements in places. I have made some suggestions in the following comments.

Method section:

Page4 lines 95-96

o The authors mentioned to “comprehensive preterm care system” in two cities in different hospital settings”. I recommend providing more details on this system and why they call it comprehensive. Also, what is the difference between a general hospital and a regional referral hospital?

Page5 line 100

o Please define “critically ill” preterm infants that you were excluded from the study.

Page5 line3 97-110

o The authors mentioned “Our study population included caregivers of preterm infants defined as parents (mothers and 99 fathers), or other relatives or carers.”. however they did not mention how they balance these participants in the study. Did they determine the number of participants in each group of caregivers? If carers were from hospital personnel then it could make bias during the focus group discussion.

Result section:

o I recommend making a table from the main themes/codes in the result section and mentioning the important points of each theme/code in the table.

Conclusion section:

o This part does not need to use references. Comparing the results with other studies should mention in the discussion section.

Reviewer #3: I would like to congratulate the authors for the beautiful work! This manuscript describes the experiences of 56 caregivers of pre-term babies while transition from hospital to home-care in Uganda and aims to identify the needs, opportunities and potential interventions for improving support systems. The introduction is well written; the methods, especially of conducting thematic analysis are appreciable and the results are systematic. The hard-work put by all the authors in the manuscript is commendable.

I have few comments and suggestions, which might be helpful to strengthen the manuscript:

1. Can you please explain in the methods section that how many participants were selected from each hospital? Was the sample drawn proportionate to the number of pre-term births happened at both the hospitals or just convenience sampling was done? As in the inclusion criteria, parents of 1-12 months aged infants are taken and in the SDCs table 1, there are two age- groups of infants, but caregivers of 1-2 months babies are not there, did that happen by chance?

2. For pre-testing FGD guide, how did you select the participants? It is mentioned that caregivers from different localities were selected. Was the hospital data used to identify them or some other method used?

3. Infants may have different requirements or needs at different age; for example, KMC can be crucial for younger babies than their older counterparts. So, I am curious to know that were there any different criteria/questions to interview the caregivers of different aged infants, or have you recorded any typical responses from them?

4. Initially it has been mentioned that healthcare providers were very supportive (line no. 177), but later it is stated that they were not empathetic (line no.238) and were rude. So these are contradicting statements, please explain or resolve it.

5. With due respect, if possible, please give some explanation in the introduction section about stigma of having or experiencing pre-term babies in the study population. As a reader it feels little awkward to read so many times that people felt disgusted about the pre-term baby or hated them. If some background is set explaining such behavior, it will be easier for readers from other cultures to understand.

6. The title of the study though only depicts experiences of the caregivers during the transition, but aim of the paper is broad and intends to identify potential interventions to improve support system. I have felt that the authors should elaborate the proposed interventions which are also practical in that region, some more explanation should be given towards the last part of discussion section which justifies the aim.

7. Can you give a reference to the lines 353-355 (peer-support).

Reviewer #4: (No Response)

7. PLOS authors have the option to publish the peer review history of their article (what does this mean?). If published, this will include your full peer review and any attached files.

**Do you want your identity to be public for this peer review?** For information about this choice, including consent withdrawal, please see our Privacy Policy.

Reviewer #1: No

Reviewer #2: **Yes: **Elahe Khorasani

Reviewer #3: **Yes: **Dr Reetu Passi

Reviewer #4: **Yes: **Mr Hlologelo Malatji

---

## [Decision Letter · Decision Letter 2]

17 Apr 2023

Transition from hospital to home care for preterm babies: a qualitative study of the experiences of caregivers in Uganda

PGPH-D-22-00378R2

Dear Associate Professor Nair,

We are pleased to inform you that your manuscript 'Transition from hospital to home care for preterm babies: a qualitative study of the experiences of caregivers in Uganda' has been provisionally accepted for publication in PLOS Global Public Health.

Best regards,

Julia Robinson

Executive Editor

Reviewer Comments (if any, and for reference):

Reviewer's Responses to Questions

**Comments to the Author**

1. If the authors have adequately addressed your comments raised in a previous round of review and you feel that this manuscript is now acceptable for publication, you may indicate that here to bypass the “Comments to the Author” section, enter your conflict of interest statement in the “Confidential to Editor” section, and submit your "Accept" recommendation.

Reviewer #1: All comments have been addressed

Reviewer #3: All comments have been addressed

Reviewer #4: All comments have been addressed

2. Does this manuscript meet PLOS Global Public Health’s publication criteria? Is the manuscript technically sound, and do the data support the conclusions? The manuscript must describe methodologically and ethically rigorous research with conclusions that are appropriately drawn based on the data presented.

Reviewer #1: Yes

Reviewer #3: Yes

Reviewer #4: Yes

3. Has the statistical analysis been performed appropriately and rigorously?

Reviewer #1: N/A

Reviewer #3: Yes

Reviewer #4: N/A

4. Have the authors made all data underlying the findings in their manuscript fully available (please refer to the Data Availability Statement at the start of the manuscript PDF file)?

Reviewer #1: Yes

Reviewer #3: Yes

Reviewer #4: Yes

5. Is the manuscript presented in an intelligible fashion and written in standard English?

Reviewer #1: Yes

Reviewer #3: Yes

Reviewer #4: Yes

6. Review Comments to the Author

Reviewer #1: (No Response)

Reviewer #3: All the comments are addressed and I do not have any further queries. The editor can take decision. Thank you.

Reviewer #4: The authors have adequately addressed my comments. The manuscript has significantly improved and read better.

7. PLOS authors have the option to publish the peer review history of their article (what does this mean?). If published, this will include your full peer review and any attached files.

**Do you want your identity to be public for this peer review?** For information about this choice, including consent withdrawal, please see our Privacy Policy.

Reviewer #1: No

Reviewer #3: **Yes: **Dr Reetu Passi

Reviewer #4: **Yes: **Hlologelo Malatji
